# Behavioral and Pharmacokinetics Studies of *N*-Methyl-2-Aminoindane (NM2AI) in Mice: An Aminoindane Briefly Used in the Illicit Drug Market

**DOI:** 10.3390/ijms24031882

**Published:** 2023-01-18

**Authors:** Micaela Tirri, Giorgia Corli, Raffaella Arfè, Beatrice Marchetti, Sabrine Bilel, Tatiana Bernardi, Federica Boccuto, Sara Odoardi, Serena Mestria, Sabina Strano-Rossi, Matteo Marti

**Affiliations:** 1Department of Translational Medicine, Section of Legal Medicine and LTTA Center, University of Ferrara, 44121 Ferrara, Italy; 2Department of Environmental Sciences and Prevention, University of Ferrara, 44121 Ferrara, Italy; 3Forensic Toxicology Laboratory, Department of Health Surveillance and Bioethics, Università Cattolica del Sacro Cuore F. Policlinico Gemelli IRCCS, 00169 Rome, Italy; 4Collaborative Center for the Italian National Early Warning System, Department of Anti-Drug Policies, Presidency of the Council of Ministers, 00186 Rome, Italy

**Keywords:** NM2AI, novel psychoactive substances, aminoindanes, stimulant, hallucinogen

## Abstract

Drug forums are considered as the main platform sources that have contributed to the increase in NPS popularity, especially for those not yet known to law enforcement and therefore not yet illegal. An example is the new synthetic stimulant NM2AI, which has a very short history of human use and abuse. Little is known regarding this compound, but some information from internet forums and the scientific literature indicates NM2AI as a structural derivate of MDAI, which is known for its entactogenic activity. Indeed, the purpose of this study is to evaluate, for the first time, the in vivo acute effect induced by the intraperitoneal injection of NM2AI (1–10–30–100 mg/kg) in mice. We demonstrate the sensory (by visual placing and object tests) and physiological (core temperature measurement) function variations, nociceptor (by tail pinch test) and strength (grip test) alterations, and sensorimotor (time on rod and mobility) decrease. Moreover, we verify the mild hallucinogenic effect of NM2AI (by startle/prepulse inhibition test). Lastly, we perform a pharmacokinetic study on mice blood samples, highlighting that the main active metabolite of NM2AI is 2-aminoindane (2AI). Taken together, our data confirm the suspected entactogenic activity of NM2AI; however, these in vivo effects appear atypical and less intense with respect to those induced by the classic stimulants, in surprising analogy with what is reported by networked users.

## 1. Introduction

The introduction of new psychoactive substances derived from structures of already known substances in the classic drug market is a phenomenon that has varied over the years according to drug use patterns in humans. However, new chemical structures continue to appear every year in order to avoid legal restrictions and consequences [1,2]. According to the European Monitoring Centre of Drugs and Drug Addiction (EMCDDA), in 2020, almost 7 tons of NPS were seized and, among these, the most abundant types were synthetic cannabinoids and cathinones [3]. However, the dynamic nature of the drug market makes real quantification almost impossible. This is primarily due to the fact that drug habit trends seem to vary every year, especially in the last two years: the advent of COVID-19 and the restrictions imposed by government authorities have favored the introduction of low tablet dosages, apparently marketed as more suitable for domestic consumption [3]. Secondly, many substances appear and disappear rapidly, posing serious challenges for their monitoring. This is the case for *N*-methyl-2-aminoindane (NM2AI), a synthetic aminoindane of which there is a lack of information in the published scientific literature. It was detected in Europe, from drugs seized by police authorities, and analyzed in an Italian forensic laboratory in the period between 2013 and 2015 [4]. After this scientific investigation, NM2AI seemed to disappear from the illicit drug market, as confirmed by a snapshot of online forum discussions that show that its brief monitoring dates back only to 2014 [5]. This was until 2021, when a DEA alert was issued in May regarding the confirmed presence and quantification of NM2AI in six urine samples from California and Minnesota [6]. In addition, a recent study conducted by Mestria and her colleagues reported the names, structures, exact protonated masses, and retention times of the main NM2AI metabolites [7]. Unlike the published scientific literature, the psychonaut web page and forum (Bluelight.com, Psychonautwiki and many others) contain grey information sources on NM2AI, including data, related chemistry, pharmacology, dosage of use, subjective desired and/or adverse effects (with relative onset and total duration), and possible interactions with other medicinal products or illicit drugs.

*N*-methyl-2-AI (NM2AI) is an *N*-methylated derivative of 2-aminoindane (2AI), analogous to amphetamine. Both, in turn, are derivatives of the controlled analog 5,6-methylenedioxy-2-aminoindane (MDAI), which has a methylenedioxy substituent that 2AI and NM2AI do not have (Figure 1).

MDAI has become a popular club drug [8], considered a good alternative to MDMA by users, and, for this reason, it has become the fifth most sold substance on the illicit drug market [9]. In fact, the desired effects of MDAI for which it is commonly sought are similar to those of MDMA: increase in empathy and sociability, mild euphoria, and sexual desire [10,11]. Even the monoamine-releasing properties of MDAI are comparable to those of MDMA: it causes non-neurotoxic 5-HT release acting with mild empathogenic properties [12,13] but also inhibits the uptake of DA, NE, and 5HT [14]. Owing to this MDMA-like profile, MDAI is considered a typical entactogen substance. Similarly, MDAI produces both stimulant and hallucinogenic effects, as with MDMA [12,15,16]. Despite this, due to the lack of research regarding the analogous 2AI and NM2AI, all discussion regarding the pharmacy–toxicology effects of these drugs is purely based upon their structural similarities to other stimulants. Similar to MDAI, 2AI and NM2AI also have stimulating effects, as reported in various experiences on internet forums and on Psychonautwiki, in which users describe in detail their subjective experiences following the administration of a determinant dose of NM2AI. These forums report that the effects are somewhat similar but less intense than those induced by MDMA, while being stronger than those induced by caffeine or Modafinil (a stimulant of the central nervous system that has given particularly promising results in psychostimulant replacement therapy, with the advantage of a lower abuse risk) [17]. Therefore, online psychonauts have recommended NM2AI use for some physical activities, from home cleaning to running or for dancing in clubs. Among the most commonly sought-after effects are an improvement in motivation, capacity for analysis and memory, acceleration of thought, awakening, and euphoria. The onset of the desired effects is reported to occur within 10–30 min (2AI) and 30–60 min (NM2AI), with an overall duration lasting 1–4 h. However, among the most common side effects, users have reported an increase in cardiac activity, hyperthermia and sweating (in fact, it is often advised by users to remember to drink), facial spasms, erectile dysfunction, and the so-called “comedown” effects. This latter is characterized by anxiety, irritability, cognitive fatigue, demotivation, deceleration of thinking, and inability to fall asleep. Users also report desired effects in relation to the different taken doses, distinguishing these as “threshold”, “light”, “common”, and “strong”. The “common” taken dosage of 2AI is 10–20 mg, while lower dosages are labelled as “light” (5–10 mg) or “threshold” (3–5 mg) and a dose between 20 and 40 mg is considered “strong”. On the other hand, users report that with an “up to 150–200 mg” dosage of NM2AI (classified as “strong”), there is a greater likelihood of incurring side effects. However, the recommended dose of NM2AI is between 100 and 150 mg, while lower dosages are considered “light” (50–100 mg) or “threshold” (>50 mg) [17]. Furthermore, some other information that was provided in an online forum in 2015 stated that NM2AI was used mainly during parties as an alternative to MDAI [18] and to achieve the desired stimulant effects as produced by the classic amphetamines, such as MDMA [19].

Despite its brief history of human consumption as a recreational drug [17,19], the present study aims at filling the gap regarding the lack of pharmaco-toxicological and pharmacokinetic information about NM2AI. Thus, we investigated the acute systemic administration of a light and threshold, strong, and heavy dosage range: 1–10–30 and 100 mg/kg. The purpose of this study was to evaluate the in vivo effects of NM2AI injection (1–10–30–100 mg/kg, i.p.) on the most important sensorial (visual) and motor (spontaneous and inducted activity) behavioral changes, the alteration of core temperature, the pain mechanical threshold, and muscle strength. Moreover, given the remarkable chemical similarity to other illegal substances already characterized and identified for their hallucinogenic action (MDMA in mice and MDAI in rats) [15,20], we considered it appropriate to investigate the effect of NM2AI also on startle/prepulse inhibition (PPI) responses in mice. Lastly, we evaluated also the pharmacokinetic profile of NM2AI (10 mg/kg, i.p.) in blood samples.

## 2. Results

### 2.1. Behavioral Studies

Mice did not appear to show effects of variation from the control groups (data not shown). Similarly, the vehicle’s administration did not vary the visual responses of the animals during the five hours of the experiment (Figure 2, Figure 3 and Figure 4).

#### 2.1.1. Evaluation of the Visual Response

Acute systemic administration of NM2AI (1–10–30–100 mg/kg, i.p.) varied both the visual object (Figure 2A) and placing (Figure 2B) responses in mice during the 5 h observations. NM2AI significantly reduced the visual object response in mice only at high doses (10–30 and 100 and mg/kg). Additionally, the most marked inhibition was visible after 10 mg/kg administration, and the effect persisted only for three hours and then almost returned to baseline values ((Figure 2A; ANOVA, main effect of treatment (F_(4,280)_ = 40.85 and *p* < 0.0001), time (F_(7,280)_ = 19.77 and *p* < 0.0001), and time x treatment interaction (F_(28,280)_ = 4.953 and *p* < 0.0001)). NM2AI reduced also the visual placing response in a dose-dependent manner. NM2AI was effective already at 15 min after injection at all dosages tested, but the effect persisted for up to 5 h only with the highest dose (100 mg/kg) ((Figure 2B; ANOVA, main effect of treatment (F_(4,280)_ = 80.49 and *p* < 0.0001), time (F_(7,280)_ = 36.30 and *p* < 0.0001), and time x treatment interaction (F_(28,280)_ = 4.350 and *p* < 0.0001)).

#### 2.1.2. Evaluation of Acoustic Response

Over the 5-h observational period, the acoustic response did not change in the vehicle groups. The acute systemic administration of NM2AI did not alter the acoustic response at any dosage (1–10–30–100 mg/kg, i.p.).

#### 2.1.3. Evaluation of Tactile Response

Over the 5-h observational period, the overall tactile response did not change in the vehicle groups. The acute systemic administration of NM2AI did not alter the tactile responses of mice at any dosage (1–10–30–100 mg/kg, i.p.).

#### 2.1.4. Evaluation of Body Temperature

Acute systemic administration of NM2AI (1–10–30–100 mg/kg, i.p.) did not vary the core body temperature of the animals during the five hours of the experiment (Figure 3A). In contrast, 100 mg/kg of NM2AI induced in mice a significant reduction in core body temperature from 85 min after injection, and the effect persisted for up to 5 h ((Figure 3A; ANOVA, main effect of treatment (F_(4,245)_ = 46.81 and *p* < 0.0001), time (F_(6.245)_ = 5.289 and *p* < 0.0001), and time x treatment interaction (F_(24,245)_ = 2.079 and *p* = 0.0030)).

On the other hand, the surface temperature did not change over the 5-h observational period, in vehicle groups and also in the group of mice treated with NM2AI (1–10–30–100 mg/kg, i.p.).

#### 2.1.5. Evaluation of Pain Induced by Two Stimuli: Mechanical (Tail Pinch Test) and Thermal (Tail Withdrawal)

Acute systemic administration of NM2AI (1–10–30–100 mg/kg, i.p.) impaired the mice’s response to the tail pinch test in a dose-dependent manner during the 5-h observation (Figure 3B). In particular, the administration of lower doses of NM2AI (1 and 10 mg/kg i.p) did not induce mechanical analgesia in mice, while intermediate and high dosages increased the pain threshold in mice. The increased tolerance for mechanical pain persisted for up to five (30 mg/kg) and four (100 mg/kg) hours after treatment, respectively ((Figure 3B; ANOVA, main effect of treatment (F_(4,245)_ = 109.5 and *p* < 0.0001), time (F_(6,245)_ = 16.48 and *p* < 0.0001), and time x treatment interaction (F_(24,245)_ = 3.253 and *p* < 0.0001)). In contrast to the tail pinch test, the acute systemic administration of NM2AI (1–10–30 mg/kg, i.p) did not induce any effect on the tail withdrawal response of mice over the 5 h of observations.

#### 2.1.6. Sensorimotor Activity Assessment (Accelerod and Immobility Time Test)

The acute systemic administration of NM2AI (1–10–30–100 mg/kg, i.p.) induced a dose-dependent biphasic pattern in the spontaneous locomotor activity of mice (mobility test). In particular, in the first 15 min after injection, there was a significant decrease in the animals’ exploration activity at all the doses tested, and this effect was reversed after one hour of NM2AI administration. After three hours (185 min) of injection, a delayed significant stimulant effect appeared with higher doses of NM2AI (30–100 mg/kg, i.p.) ((Figure 4A; ANOVA, main effect of treatment (F_(4,280)_ = 1.807 and *p* = 0.1276), time (F_(7,280)_ = 163.5 and *p* < 0.0001), and time x treatment interaction (F_(28,280)_ = 11.01 and *p* < 0.0001)).

On the other hand, acute systemic administration of NM2AI (1–10–30–100 mg/kg, i.p.) acted differently on the dynamic conditions of movement in relation to the dose administered under motor stimulation conditions verified in the accelerod test. In particular, the lowest dose (1 mg/kg) seemed to be ineffective, and only the intermediate dose (10 mg/kg) induced a hyper-stimulating action in animals, with an increase in the time spent on the rod. Higher doses (30 and 100 mg/kg) induced instead a long-lasting inhibition of mice’s movement abilities on the rod ((Figure 4B; ANOVA, main effect of treatment (F_(4,280)_ = 140.6 and *p* < 0.0001), time (F_(7,280)_ = 4.002 and *p* = 0.0003), and time x treatment interaction (F_(28,280)_ = 3.820 and *p* < 0.0001)).

#### 2.1.7. Evaluation of Skeletal Muscle Strength (Grip Strength Test)

Acute systemic administration of NM2AI (1–10–30–100 mg/kg, i.p.) induced a dose-dependent inhibition of skeletal muscle strength (Figure 4C). In particular, the lowest dose tested (1 mg/kg i.p.) did not affect the muscle strength of mice. However, the intermediate (10 and 30 mg/kg) and higher (100 mg/kg) doses reduced significantly the skeletal muscle strength, and the effect persisted for up to four hours of observation ((Figure 4C; ANOVA, main effect of treatment (F_(4,280)_ = 85.34 and *p* < 0.0001), time (F_(7,280)_ = 14.93 and *p* < 0.0001), and time x treatment interaction (F_(28,280)_ = 3.803 and *p* < 0.0001)).

### 2.2. Startle/Prepulse Inhibition Analysis

Vehicle injection did not change the startle/PPI response in mice (Figure 5).

Acute systemic administration of NM2AI (1–10–30 mg/kg, i.p.) induced a variation in the mice’s startle/PPI responses. In fact, the *T*-test detected a significant effect on mice’s startle amplitude with respect to the vehicle only at a higher dose (30 mg/kg, i.p.; Figure 5A; effect of treatment: t = 2.734, Df = 16 and *p* = 0.0147). At the same time, one-way ANOVA detected also a significant effect on startle amplitude between the three doses tested (Figure 5A; F_(2,24)_ = 7.201 and *p* = 0.0036). On the other hand, the *T*-test detected a significant effect of mice’s PPI responses with respect to the vehicle at different prepulse intensities: 68 dB at 1 mg/kg (Figure 5B; effect of treatment: t = 8.849, Df = 16 and *p* < 0.0001) and 30 mg/kg (Figure 5B; effect of treatment: t = 2.684, Df = 16 and *p* = 0.0163); 75 dB at 1 mg/kg (Figure 5B; effect of treatment: t = 5.925, Df = 16 and *p* < 0.0001) and 30 mg/kg (Figure 5B; effect of treatment: t = 3.437, Df = 16 and *p* < 0.0001); 85 dB at 1 mg/kg (Figure 5B; effect of treatment: t = 6.761, Df = 16 and *p* < 0.0001) and 30 mg/kg (Figure 5B; effect of treatment: t = 3.426, Df = 16 and *p* = 0.0035). At the same time, one-way ANOVA detected also a significant effect on mice’s PPI responses between the three doses at each intensity tested (Figure 5B): 68 dB (F_(2,24)_ = 6.00 and *p* = 0.0077), 75 dB (F_(2,24)_ = 8.267 and *p* = 0.0019), and 85 dB (F_(2,24)_ = 6.430 and *p* = 0.0058).

### 2.3. Pharmacokinetic Study

NM2AI’s Cmax in blood samples was reached at 30 min from injection and ranged from 0.9 to 3.7 µg/mL (average 2.7 µg/mL). NM2AI’s mean concentration in blood sampled at 120 min post-injection was 1.7 µg/mL, that at 240 min was 0.71 µg/mL, and that at 300 min was 0.45 µg/mL. The main active metabolite, 2-aminoindane (2AI), was present in all the blood samples analyzed, with mean concentrations of 71, 57, 37, and 20 ng/mL at 30, 120, 240, and 300 min, respectively. Metabolite to parent drug ratios, 2AI/NM2AI, increased with the sampling time for all the mice. These findings are depicted in Figure 6.

## 3. Discussion

Relying on recent pharmacokinetics studies that had already discovered the levels of NM2AI and its main metabolites in mice urine, blood, and fur [7], this study revealed, for the first time, the in vivo pharmaco-toxicological effects of NM2AI. While the majority of ring-substituted aminoindane derivatives are already classified as monoamine transporter substrates, with relevant affinity for adrenergic, dopaminergic, and serotonergic receptors [21,22,23,24], data on the non-ring-substituted aminoindanes, such as 2-aminoindane (2-AI) and NM2AI, are very scarce. Moreover, despite the fact that NM2AI is widely recognized as a low-power compound when compared to well-known stimulant and/or entactogen molecules, several online sources still bear witness to its use [17,19]. Therefore, we showed that the acute systemic administration of NM2AI (1–10–30–100 mg/kg) induces a variation in the core temperature, reduces mice’s sensorimotor responses to visual (placing and object) stimuli, impairs spontaneous and stimulated motor activity, reduces skeletal muscle strength, increases mechanical analgesia, and impairs the sensorimotor gating ability in mice.

### 3.1. Visual Responses

The different efficacy of NM2AI in producing behavioral effects in mice, affecting the variation in responses to visual stimuli (Figure 2A,B) but not responses to other stimuli (tactile and acoustic test; data not shown), is worth noting. This result could be related to the typical rigid conformation structure of aminoindane derivatives, in which the α-carbon is directly connected to the ortho-position of the aromatic ring (2-aminoindan) [25]. Indeed, this chemical structure facilitates the primary activity of the serotonin (5-HT) system, although stimulation was limited [26]. The serotonin receptor system consists of a wide family of subtypes, which are widely distributed in both the human central nervous system [27] and other systems and organs, such as the gastrointestinal (GI) tract [28], skeletal muscle, or eyes, in which 5-HT is a modulator of the retina [29]. In line with the literature [30], in our study, the visual object response’s decrease may be due to an impairment of serotoninergic transmission to the retina. However, there are many examples of different MDMA-like compounds, such as methiopropamine (MPA) [31], and hallucinogenic [32,33,34] and dissociative anesthetic [35] drugs that inhibit responses in mice and provoke sensory alterations [36], especially visual ones [15]. This could be attributed to their selectivity for 5HT_2A_ in cortical brain areas [37]. In fact, the genetic inactivation of the receptor 5HT_2A_ blocks the behavioral effects of many hallucinogenic compounds in both humans and laboratory animals [38]. Moreover, it is relevant to note that NM2AI has altered the visual response to placing more deeply than the object test in mice (Figure 2A,B). This may be due to the vestibular spinal pathway, which allows the mouse to properly coordinate visual information (approach to the floor) with its muscle extension movement to prepare for contact with the ground [39]. Manier and colleagues highlighted that NM2AI selectively inhibits NET and DAT, causing an outflow of NE and DA and leading to their accumulation in the synaptic fissure [40]. In agreement with our previous study [31], this evidence may explain why NM2AI by NE release inhibits more deeply the mice’s visual placing responses with respect to the object one: it cause a vestibule–ocular reflex variation, which provides simultaneous body postural control and the optokinetic response of the forelimbs through β- and α2-receptor activation [41]. Taken together, these findings suggest that NM2AI could act by promoting both cortical and vestibular visual circuit alteration through the direct activation of the 5HT_2A_ receptor and/or indirect sympathomimetic activation of the spinal pathway.

### 3.2. Core Body Temperature

It is well established that the use of amphetamine derivatives can lead to serious complications related to serotonin syndrome and toxicity [20,42,43,44]. In fact, aminoindanes such as MDAI influence the release of 5-HT and NE in a manner comparable to MDMA [45]. As a result, many subjective effects of MDAI are reported to be very similar to those of MDMA [8] and many studies have already reported MDMA-induced temperature changes in rodents [46,47,48]. Likewise, some cannabinoids with an indole-based structure have been associated with serious medical complications, such as psychosis, hypothermia, and catatonia, as a result of their consumption. In fact, studies show that positive allosteric modulation of the 5-HT1_A_ receptor could lead to their activation in rodents and contributes to the manifestation of hypothermia [49,50]. Taken together, the results of this study show that the hypothermic effects of aminoindane derivates are mediated by 5-HT release and therefore are in accordance with our data obtained. Moreover, mild hypothermia caused by NM2AI could be due to NE release. Several studies have shown that the systemic administration of NE and DA induces hypothermia in mice [51,52,53] and aminoindanes such as 5-IAI, MDAI, and 2-AI preferentially inhibit SERT and NET, increasing the related monoamine efflux [45].

### 3.3. Analgesic Profile Responses

Aminoindanes have been studied also for their analgesic effects and ability to increase blood pressure and respiration [54,55]. Interestingly, NM2AI does not appear to have analgesic effects on humans, unlike the effects reported by many users on blogs for its analog 2AI (which even describe them as being similar to opioids) [56]. It is not clear whether this is due to an individual factor related to the co-administration of other drugs by the user or simply to the chosen dosage to be tested. This evidence is in agreement with opinions and advice found in forums, which confirm that the effects of NM2AI can progressively intensify proportionally to the dosage [17]. Compared to information reported by psychonauts, it is interesting to note that mice respond differently to pain stimuli tests. In fact, following the administration of NM2AI, the mice’s responses to thermal pain seemed to be unaffected, but the drug appeared to increase the mechanical pain resistance to the tail pinch test, thus suggesting the inherent analgesic capacity of the molecule. This result could be due to the differences in sensitivity to neuropathic pain. However, the increased tolerance to mechanical pain could be due to the facilitation of NE release following the administration of the substance. In fact, a study by Simmler et al. confirmed that aminoindanes such as 2-AI selectively inhibit NET and increase the release of NE, and its profile is relatively similar to benzodiazepine [22]. In addition, in a different study, it has been demonstrated that serotonin–norepinephrine reuptake inhibitors (SNRIs) effectively reduce pain-related behavior in an animal model [57]. Consistently, our results for the in vivo pharmacokinetic study on mouse blood samples showed that the 2AI/NM2AI ratio increased slightly over time. Therefore, bioactivation to 2AI, although in a slightly higher quantity than NM2AI, may still be directly responsible for the analgesic effect shown in the tail pinch test.

### 3.4. Motor Activity and Grip Strength

Contrary to what is expected for a cathinone such as mephedrone [58], in humans and in rodents; methylone and butylone [59], or stimulants such as α-PVP, MDPV [31,60], and MDMA [15], which increase locomotor activity in a dose-dependent manner; as well as amphetamine, which increases forelimb grip strength [61], NM2AI seemed to result in a different response that depended on the test performed and the dose tested. In particular, we have demonstrated, using the accelerod test, that only the dose of 10 mg/kg of NM2AI is able to produce the facilitation of locomotion in mice. On the other hand, NM2AI produced a biphasic profile in the mobility test. This effect seemed to be similar to those induced by MDAI and MDMA in mice [14], characterized by a brief decrease in spontaneous locomotion followed by the prolonged facilitation of mice’s activity. It is important to note that the increase in motor activity in vivo occurs precisely in correspondence to the increase in blood levels of 2AI. Moreover, the forelimb grip muscle strength in mice decreased as a corresponding result of decreased motor activity. Taken together, this effect highlights once again the importance of the structure–activity correlation of NPS. In fact, 2AI is a dopaminergic and noradrenergic stimulating drug rather than a serotonin (5-HT) releasing agent, differently to 5-IAI, which stimulates mostly serotonin (5-HT) and slightly dopamine (DA), releasing, to a lesser extent, norepinephrine (NE) [22]. These findings show that even a minor change in the basic molecular structure (aminoindanes) can cause different effects; indeed, NM2AI could preferentially bind to the serotoninergic receptor system over the dopaminergic receptor system, typical of different stimulants, suggesting empathogen-like effects [14]. This may also explain why the dose range to achieve a stimulating effect of NM2AI is high (to increase DA release), or why users consume often these drugs in cocktails with other stimulants, such as MDMA, cocaine, or MXE, a dissociative compound (https://psychonautwiki.org/wiki/NM-2-AI; accessed on 2 December 2022). However, there have already been fatalities related to 5-IAI and 2-AI [62]. Thus, if users consume it in a cocktail of drugs, its effects will be likely more harmful and lethal, causing neuro- and cardiotoxicity [8].

### 3.5. PPI Effect

Given that NM2AI did not impair the acoustic responses of mice, we investigated through PPI whether it could, however, impair the ability of mice to integrate and process acoustic information, which could translate into and explain the cognitive deficit reported in forums by users [17]. Previous studies highlighted that MDAI’s disruptive effects on sensorimotor gating in rats were clearly evident [20] and MDMA in mice reduced sensorimotor gating at multiple prepulse intensities [15]. Therefore, owing to the resemblance of NM2AI to MDMA-like compounds, which typically inhibit PPI, a similar disruption was predicted. In fact, we clearly proved that NM2AI not only disrupted the ability of mice to receive visual information and altered their spontaneous activity, but also impaired its integration, and its effects are similar to those of hallucinogens [33,34].

## 4. Materials and Methods

### 4.1. Animals

Male ICR (CD-1^®^) mice, weighing approximately 30–35 g (Centralized Preclinical Research Laboratory, University of Ferrara, Italy), were group-housed (5 mice per cage; floor area per animal was 80 cm^2^; minimum enclosure height was 12 cm) under a 12:12-h light–dark cycle (light on 6:30 a.m.) with standard temperature (20–22 °C), humidity (45–55%) and ad libitum access to food (Diet 4RF25 GLP; Mucedola, Settimo Milanese, Milan, Italy) and water. The experimental protocols performed in the present study were in accordance with the U.K. Animals (Scientific Procedures) Act of 1986 and associated guidelines and the new European Communities Council Directive of September 2010 (2010/63/EU). Experimental protocols were approved by the Italian Ministry of Health (license n. 335/2016-PR) and by the Animal Welfare Body of the University of Ferrara. According to the ARRIVE guidelines, in order to minimize the number of animals used, but especially the pain and discomfort of the test subjects, for the overall study, 40 mice were used.

### 4.2. Drug Preparation and Dose Selection

NM2AI was obtained through police seizure on Italian soil by law enforcement agencies and purchased from LGC Standards S.r.L. (Milan, Italy). The range doses used in experiments were chosen based on the literature [63], on the basis of previous preliminary studies in rodents [15], and on reported behavioral and neurological effects in humans [17]. Indeed, considering the human experiences with NM2AI, typical recreational doses (100–150 mg) produce the pleasant effects previously reported, while the lower dose of NM2AI (50–100 mg) is considered a mild dosage that causes few effects. For this reason, we chose 1–10–30–100 mg/kg as the range of doses to be tested. In fact, with reference to the interspecies scale [64], a dose of 1 mg/kg drug in a mouse is equivalent to a dose of ~0.081 mg/kg (~5.67 mg) taken by a human weighing 70 kg. Thus, a human dose of ~10 mg/kg is equivalent to ~0.813 mg/kg (~56 mg), and a human dose of 30 mg/kg is equivalent to ~2.439 mg/kg (~170 mg), while a human dose of 100 mg/kg is equivalent to ~8.13 mg/kg (~569 mg). NM2AI was dissolved in saline solution and administered by intraperitoneal injection to mice (volume 4 μL/g). 

### 4.3. Experimental Protocol of Behavioral Studies

The effect caused by the injection of NM2AI (1–10–30–100 mg/kg; i.p.) was assessed through the “safety–pharmacology” system, a battery of different behavioral tests widely used in our laboratory. Animals’ behavior was evaluated between 8:30 and 2:00 p.m. in a blinded manner, by trained observers working in pairs, while being filmed and analyzed off-line by a different trained operator [65]. Voluntary and involuntary sensorimotor responses were assessed in a consecutive manner according to the following sequence: observation of visual placing and object (frontal and lateral view) responses, determination of mechanical (tail pinch) acute pain, motor activity (mobility time and rotarod), and the evaluation of skeletal muscle strength (grip test).

NM2AI (1–10–30–100 mg/kg; i.p.) or saline was administered 5 min before starting laboratory tests [15].

In the following, we provide a detailed description of the execution tests that showed, in the results, a change in the animals’ behavioral activity.

#### 4.3.1. Evaluation of the Visual Responses (Visual Object and Placing Test)

The visual response is assessed by two tests investigating whether the mouse’s ability to capture visual information, either when the animal is stationary (visual response of the object) or moving (visual response of positioning), may vary with the administration of a substance.

The visual object response test is to bring a small dentist’s mirror close to the mouse’s field of view in a horizontal arc or at the side of the body, and then cause it to move or turn or withdraw its head [66]. The procedure was conducted bilaterally and the maneuver was repeated 3 times. For the frontal visual response, a white horizontal bar was moved frontally to the mouse head and this was also repeated 3 times. The score assigned was 1 if there was a reflection in the mouse’s movement or 0 if not. The total value was calculated by adding the scores obtained in both visual response tests (overall score 9). Evaluation of the visual object response was performed at 0, 10, 30, 60, 120, 180, 240, and 300 min after injection.

The visual placing response was tested using a modified tail suspension device capable of bringing the mouse to the floor at a constant speed of 10 cm/s while being captured by a camera [66]. The analysis of the frames made it possible to evaluate the start of the mouse’s reaction while it was near the floor, and the perpendicular distance between the eyes of mice and the floor was measured using an electronic ruler (for untreated mice, the distance was approximately 28 ± 4.7 mm). The assessment of the visual positioning response was performed at 0, 15, 35, 70, 125, 185, 245, and 305 min after injection.

#### 4.3.2. Evaluation of the Acoustic Response

In the acoustic test, four different stimuli (changed for intensities and frequencies) were produced behind the animal and the reflex response of the mouse was recorded by the operator [34]. Each different stimulus was repeated 3 times, giving a value of 1 if there was a response, or 0 if not present, for a total score of 3 for each sound [66]. The acoustic test was performed at 0, 10, 30, 60, 120, 180, 240, and 300 min after injection. The acoustic total score was calculated by adding scores obtained in the four tests (overall score 12). The background noise (approximately 40 ± 4 dB) and the sound from the instruments were measured with a digital sound level meter.

#### 4.3.3. Evaluation of Tactile Response

The tactile test involved testing the mouse’s responses to vibrissae, corneal, and pinnae stimuli reflexes. Data are expressed as the sum of the three afore-mentioned parameters [34].

Vibrissae reflex was evaluated by touching the vibrissae (right and left) with a thin hypodermic needle once per side, giving a value of 1 if there was a reflex (turning of the head to the side of touch or vibrissae movement) or 0 if not present (overall score 2). Pinna reflex was assessed by touching pavilions (left and right) with a thin hypodermic needle. First, the interior pavilions and then the external ones were tested. This test was repeated twice per side, giving a value of 1 if there was a reflex and 0 if not present (overall score 4). Corneal reflex was assessed by gently touching the cornea of the mouse with a thin hypodermic needle and evaluating the response, assigning a value of 1 if the mouse moved only the head, 2 if it only closed the eyelid, and 3 if it closed the lid and moved the head. The procedure was conducted bilaterally (overall score 6).

#### 4.3.4. Evaluation of Body Temperature (Core and Surface)

To better understand the effects of NM2AI on thermoregulation, we measured changes in the surface and core (rectal) temperature. Surface temperature was measured with an infrared thermometer. The core temperature was evaluated by a probe (1 mm diameter) that was gently inserted, after lubrication with liquid Vaseline, into the rectum of the mouse (to approximately 2 cm) and left in position until the stabilization of the temperature (approximately 10 s) [67,68]. The probe was connected to a Cole Parmer digital thermometer, model 8402. A cut-off for core body temperature was set at 22 °C (room temperature) as the lowest value reached by animals. The mouse body temperatures were measured at 0, 30, 50, 85, 140, 200, 260, and 320 min after injection.

#### 4.3.5. Evaluation of Pain Induced by Two Different Stimuli: Mechanical (Tail Pinch Test) and Thermal (Tail Withdrawal)

Acute mechanical nociception was evaluated using a rigid probe, connected to a digital dynamometer (ZP-50 N, IMADA, Jinnoshinden-cho Kanowari, Toyohashi, Japan), which was gently placed in the distal portion of the mouse tail and progressive pressure was applied [67]. When the mouse flicked its tail, the pressure was stopped, and the digital instrument recorded the maximum peak of weight supported (g/force). In order to avoid damage to mice tail tissue, a cut-off of 500 g/force was set [68]. The test was repeated three times, and the final value was calculated with the average of 3 obtained scores. Acute mechanical nociception was measured at 0, 35, 55, 90, 145, 205, 265, and 325 min after injection.

On the other hand, acute thermal nociception was evaluated by immersing the mouse tail in water at 48 °C and the time (in seconds) to which the tail resisted in water was recorded [69]. Mice, during the execution of this test, were held in a dark plastic cylinder (3 cm long and 6.3 cm in diameter) closed on the sides with plastic mesh, which allowed mice to breathe normally. A cut-off (15 s) was set to prevent tissue damage. Moreover, in this case, the test was repeated three times, and the final value was calculated with the average of 3 obtained times. It was fundamental that no signs of damage, burns, or changes in the tail positions of mice were observed after repeating the three consecutive tests. Acute thermal nociception was measured at 0, 40, 60, 95, 150, 210, 270, and 330 min after injection.

#### 4.3.6. Sensorimotor Activity Assessment (Accelerod and Immobility Time Test)

The effects induced on sensorimotor activity were evaluated by different widely validated and specific tests for different motor skills in spontaneous [15] and dynamic conditions of movement induction [65,70] in mice. The accelerod test consists of a rotating cylinder that automatically increases the speed in a constant manner (0–60 rotations/min in 5 min), on which the animals are placed and the time that mice can stand on it is measured. The accelerod test was performed at 0, 40, 65, 95, 150, 210, 270, and 330 min after injection. The mobility time test involved allowing the animal to move freely on a square plastic cage (60 × 60 cm) and using a stopwatch to test the time for which it remained in motion. Mouse activity was recorded using one camera (B/W USB day and night with varifocal Ugo Basile, Italy) and films were analyzed off-line in a blind manner by a different operator. The mobility time (in seconds) test was performed for 5 min at any time (0, 15, 35, 70, 125, 185, 245, and 305 min after injection).

#### 4.3.7. Evaluation of Skeletal Muscle Strength (Grip Strength Test)

The skeletal muscle strength (expressed in gram force; gf) of the mice was evaluated using the grip strength test [68,71]. The grip strength apparatus (ZP-50 N, IMADA) consisted of a wire grid (5 × 5 cm) connected to an isometric force transducer (dynamometer). The operator, holding the mouse by the tail, allowed the animal to grab the grid through its legs and it was then gently pulled backward until the grid was released. The mean of three measurements for each animal was calculated and the mean average force was determined. The grip strength was measured at 0, 15, 35, 70, 125, 185, 245, and 305 min after injection.

### 4.4. Startle/Prepulse Inhibition Analysis

Mice were tested for acoustic startle reactivity in startle chambers (Ugo Basile apparatus, Milan, Italy) consisting of a sound-attenuated, lighted, and ventilated enclosure holding a transparent non-restrictive Perspex^®^ cage (90 × 45 × 50 mm). A loudspeaker mounted laterally on the holder produced all acoustic stimuli. Peak and amplitudes of the startle response were detected by a loadcell. At the onset of the startling stimulus, 300-ms readings were recorded and the wave amplitude evoked by the movement of the animals’ startle response was measured. Acoustic startle test sessions consisted of startle trials (pulse-alone) and prepulse trials (prepulse + pulse). The pulse-alone trial consisted of a 40-ms 120-dB pulse. Prepulse + pulse trial sequence consisted of a 20-ms acoustic prepulse, 80-ms delay, and then a 40-ms 120-dB startle pulse (100-ms onset–onset). There was an average of 15 s (range = from 9 to 21 s) between the trials. Each startle session began with a 10-min acclimation period with 65-dB broadband white noise that was present continuously throughout the session. The test session contained 40 trials composed of pulse-alone and prepulse + pulse trials (with three different prepulses of 68-dB, 75-dB, and 85-dB) presented in a pseudorandomized order. Animals were placed in the startle chambers 5 min after treatment. The entire PPI test lasted 20 min. The amount of prepulse inhibition (PPI) was expressed as the percentage decrease in the amplitude of the startle reactivity caused by the presentation of the prepulse (% PPI).

NM2AI (1–10–30–100 mg/kg) was administered by intraperitoneal injection and startle/PPI responses were recorded 15 and 120 min (including the 10-min acclimation period) after drug injection.

### 4.5. Pharmacokinetic Study

In an attempt to correlate the in vivo pharmacological effects of NM2AI and any of its metabolites, a group of 20 mice were injected with NM2AI (10 mg/kg) alongside the behavioral test and used for the collection of blood samples. Tests and sampling were randomly and individually carried out for each animal. Blood sampling (total volume ~ 500 µL) was performed at specific time points (20 to 30 min, 120 min, 240 min, and 300 min), and blood was collected by a submandibular blood collection technique into 1 mL vials containing EDTA (4 mg/mL of blood) as a preservative and anticoagulant [72]. An equal volume of saline solution was subcutaneously injected into mice after each blood withdrawal, to maintain volume and osmotic homeostasis. Plasma from collected blood and urine samples was stored at −20 °C until analysis.

### 4.6. Statistical Analysis

In sensorimotor response experiments, data were expressed as arbitrary units (visual object response) or percentage of baseline (visual placing response, accelerod, mobility time, and grip strength). Antinociception (tail pinch test) was calculated as the percentage of maximal possible effect:EMax% = [(test − control latency)/(cut of time − control)] × 100.

Effects of different concentrations of each substance over time were analyzed by two-way ANOVA followed by Bonferroni’s test for multiple comparisons.

The amount of PPI was calculated as a percentage score for each prepulse + pulse trial type:% PPI = 100 − [(startle response for prepulse + pulse trial)/(startle response for pulse-alone trial)] × 100.

Startle magnitude was calculated as the average response to all of the pulse-alone trials. All the results were analyzed by treatment using a *T*-test. All analyses were performed using GraphPad Prism software (GraphPad Prism, Boston, MA, USA).

### 4.7. Blood Sample Analysis

#### 4.7.1. Blood Preparation

The blood samples collected from mice were analyzed following a deproteinization step, where 100 µL of blood spiked with IS amphetamine-d5 was added with 100 µL of methanol and stirred. After centrifugation at 10,000× *g*, 10 µL of the clear supernatant was injected into an LC–MS system.

Quantitation was performed using a five-point calibration curve, ranging from 0.1 to 5.0 µg/mL, prepared in blank blood.

#### 4.7.2. LC–HRMS Apparatus

The LC–HRMS system was composed of a Thermo ULTIMATE 3000 system equipped with an analytical column, the Thermo Acclaim RSLC 120 C18 (2.1 mm × 100 mm, 2.2 µm of particle size), coupled to a Thermo Single-Stage Orbitrap (Exactive) MS system, interfaced with a HESI source. The whole equipment setup and the column were from Thermo Fisher Scientific (Milan, Italy). Data were acquired in full scan mode over a mass range of 50 to 750 *m*/*z*. The instrument was operated in positive ion mode with a resolving power of 100.000 full width at half maximum (FWHM).

Detailed analytical conditions are reported elsewhere [7].

## 5. Conclusions

This study revealed that NM2AI (1–10–30–100 mg/kg) induced sensorimotor alterations and physiological variations, increased mechanical nociception, and reduced skeletal muscle strength. Moreover, NM2AI was able to disrupt the sensorimotor gating response in mice, acting mildly as a hallucinogenic compound. Taken together, our data show that the in vivo effects following the administration of NM2AI are surprisingly similar to those reported by networked users and confirm the suspected entactogenic activity of this compound. In addition, the pharmacokinetic analysis of mice blood samples highlighted that the main active metabolite of NM2AI was 2-aminoindane (2AI). Our results once again show how fundamental is the study of particular compounds such as NM2AI that are placed in the grey zone of the NPS illicit drug market.

## Figures and Tables

**Figure 1 ijms-24-01882-f001:**
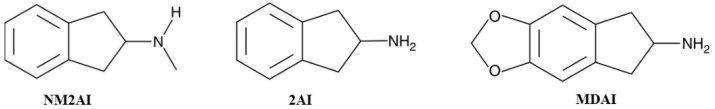
Structures of *N*-methyl-2-aminoindane (NM2AI), 2-aminoindane (2AI), and 5,6-methylenedioxy-2-aminoindane (MDAI), obtained from the Cayman Chemical website (https://www.caymanchem.com; accessed on 2 December 2022).

**Figure 2 ijms-24-01882-f002:**
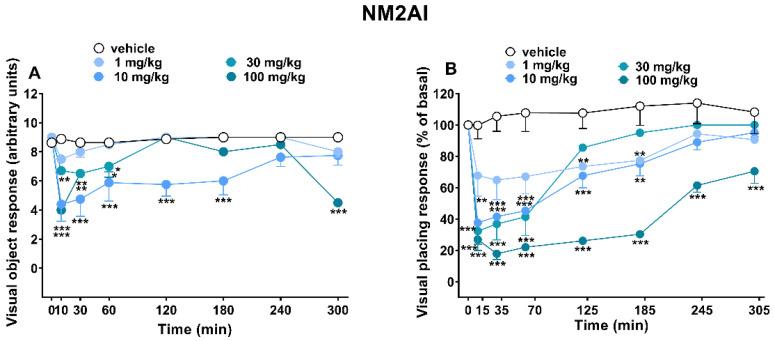
Effect of NM2AI (1–10–30–100 mg/kg; i.p.) on the visual object (**A**) and placing (**B**) response tests in mice. Data are expressed as mean ± SEM (n = 8/group). Statistical analysis was performed by two-way ANOVA followed by Bonferroni’s test for multiple comparisons for the dose–response curve of each compound at different time points (* *p* < 0.05, ** *p* < 0.01, *** *p* < 0.001 versus vehicle).

**Figure 3 ijms-24-01882-f003:**
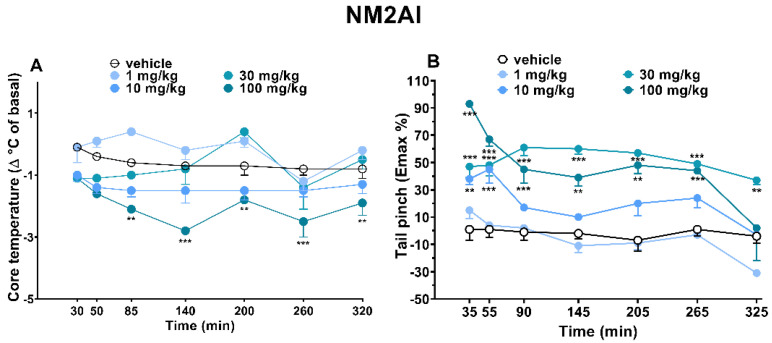
Effect of NM2AI (1–10–30–100 mg/kg; i.p.) on the core body temperature (**A**) response and tail pinch (**B**) test in mice. Data are expressed as mean ± SEM (n = 8/group). Statistical analysis was performed by two-way ANOVA followed by Bonferroni’s test for multiple comparisons for the dose–response curve of each compound at different time points (** *p* < 0.01, *** *p* < 0.001 versus vehicle).

**Figure 4 ijms-24-01882-f004:**
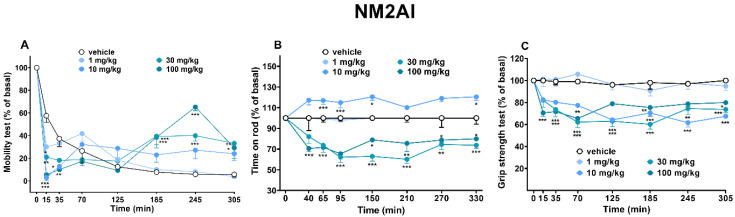
Effect of NM2AI (1–10–30–100 mg/kg; i.p.) on the mobility time (**A**), time on rod (**B**), and grip strength (**C**) response tests in mice. Data are expressed as mean ± SEM (n = 8/group). Statistical analysis was performed by two-way ANOVA followed by Bonferroni’s test for multiple comparisons for the dose–response curve of each compound at different time points (* *p* < 0.05, ** *p* < 0.01, *** *p* < 0.001 versus vehicle).

**Figure 5 ijms-24-01882-f005:**
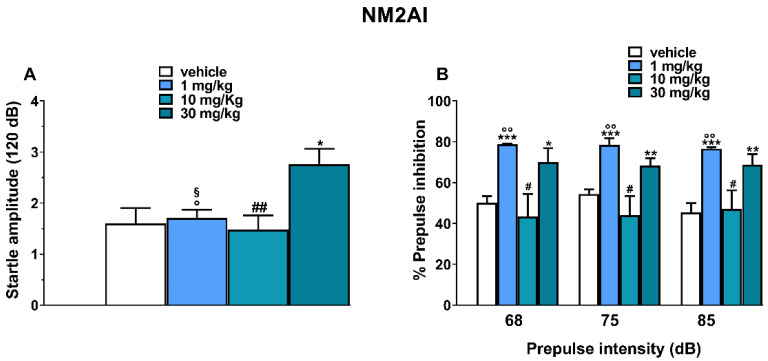
Effect of NM2AI (1–10–30 mg/kg; i.p.) on startle (**A**) and prepulse inhibition (PPI; (**B**)) in mice. For PPI, the effects are shown for three prepulse intensities (68, 75, and 85 dB) at 120 min after treatment. Prepulse inhibition (PPI) was expressed as the percentage decrease in the amplitude of the startle reactivity (data not showed) caused by presentation of the prepulse (% PPI; see Section 4) and values represent mean ± SEM of 6 animals for each treatment. The statistical analysis was performed using the unpaired *T*-test for single dosage comparison (* *p* < 0.05, ** *p* < 0.01, and *** *p* < 0.001) versus vehicle, and the one-way ANOVA followed by Tukey’s test for multiple comparisons (° *p* < 0.05 and °° *p* < 0.01 for 1 mg/kg versus 10 mg/kg; # *p* < 0.05 and ## *p* < 0.01 for 10 mg/kg versus 30 mg/kg; § *p* < 0.05 for 1 mg/kg versus 30 mg/kg).

**Figure 6 ijms-24-01882-f006:**
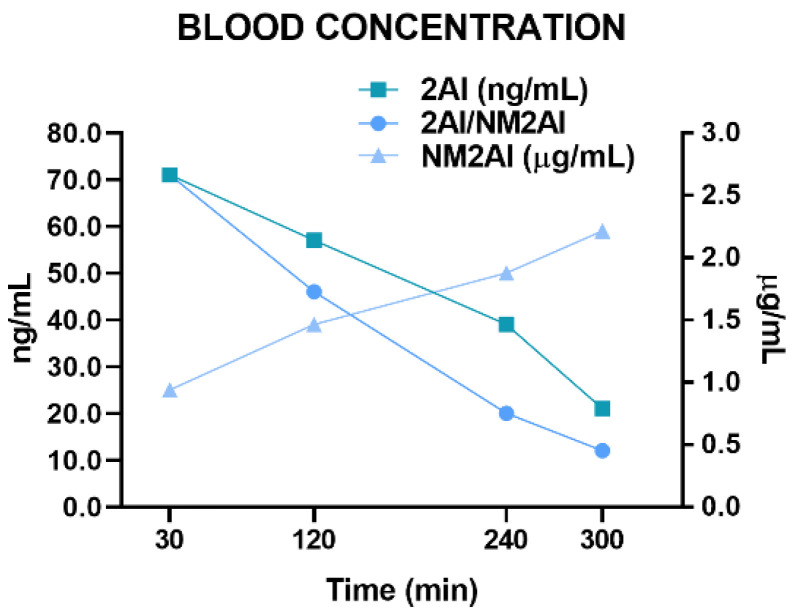
Average concentrations for NM2AI and 2AI in mice blood and average ratio of 2AI/NM2AI for different sampling times.

## Data Availability

The data presented in this study are available on request from the first (Micaela Tirri) and corresponding authors (Matteo Marti) for researchers of academic institutes who meet the criteria for access to the confidential data.

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
