# Peer review of "Behavioral and Pharmacokinetics Studies of N-Methyl-2-Aminoindane (NM2AI) in Mice: An Aminoindane Briefly Used in the Illicit Drug Market"

_ijms, 2023, doi:10.3390/ijms24031882_

Round 1
Reviewer 1 Report
1- Authors must be given an important task of their article in the title.
2- The last sentence of abstract section not clear, it must be re-write to become clear for reader.
3- Authors should be checked all symbols in the whole article.
4- In introduction section from line 84 to line 104 need to become more clarified.
5- Figures 2-to-6 need to become more clear.
6- The sub-sections of 2-1 and 4.3 have the same title, check this.
7- Title of 5- Conclusions must be corrected to conclusion.
Author Response
Response to Reviewer 1
We thank the Reviewer for his/her evaluation of our manuscript and for helpful concerns to improve the article.
In this revised version of the work we have addressed the major concerns of the referee (the parts highlighted in yellow have been included in the manuscript).
Rev1Q1: Authors must be given an important task of their article in the title.
AA: We thank the Reviewer for his/her evaluation. The title is now changed as suggested by the reviewer
Rev1Q2: The last sentence of abstract section not clear, it must be re-write to become clear for reader.
AA: We thank the Reviewer 1 for pointing out this inaccuracy. The abstract has been revised and the mentioned sentence is now corrected.
Rev1Q3: Authors should be checked all symbols in the whole article.
AA: We thank the Reviewer 1 for pointing out this inaccuracy and we have checked and corrected all symbols in the whole manuscript.
Rev1Q4: In introduction section from line 84 to line 104 need to become more clarified.
AA: We agree with the Reviewer 1 and we have clarified this section of the introduction.
Rev1Q5: Figures 2-to-6 need to become more clear.
AA: We thank the Reviewer for his/her for pointing out this this inaccuracy We have increased the axes and the clearance of all the figures.
Rev1Q6: The sub-sections of 2-1 and 4.3 have the same title, check this.
AA: We thank the Reviewer 1 for pointing out this inaccuracy and we have corrected the title of the sub-section 4.3.
Rev1Q7: Title of 5- Conclusions must be corrected to conclusion.
AA: We thank the Reviewer 1 for pointing out this inaccuracy and we have corrected the title of the section 5.
Reviewer 2 Report
I reviewed a manuscript examining the effects of a novel psychoactive compound in mice. NM2AI is pretty much dead, I think it literally died around 2016-2017 and it could be hard to purchase it on the internet, but considering that some compounds were already dead and then rose from their graves, I think it's worth publishing any data on it considering PubMed only has 4 articles on NM2AI. So, here's the list of my "buts":
1. Lines 21-22 - how is this even possible? The stuff is pretty much dead, I searched several forums and couldn't find any comments other than that NM2AI doesn't even come close to having the same effects as MDAI. I've also never seen a comparison of NM2AI to MDMA, and I study them daily. I've never seen an NM2AI post on an MDMA forum. I'd suggest changing the sentence to inform readers that the substance was developed as an analogue to MDAI, which dealers hope can work the same as MDMA.
2. You can't use an EMCDDA report from 2015 to claim that NPS are on the rise. In fact, they aren't anymore. It's all about psychedelics nowadays, and NPS are currently a niche for "drug freaks". Please review this statement. They're a public health threat, but to say that they're constantly on the rise is an exaggeration. For example, over 200 tonnes of cocaine were seized in Europe in 2020 (official EMCDDA data). If you look at the latest data, you'll see that the numbers for NPS are constantly decreasing, while the "classic" drugs are increasing.
3. 103-105 - You say that it's quite obvious that NM2AI is used at parties just like MDMA, and you quote UN data from 2015. All these statements need to go away. Just say it's an NPS, but right now it's not a big issue. However, since there is no scientific data available on NM2AI, you have decided to publish your manuscript now in case the substance resurfaces or for whatever reason. You don't have to try to prove NM2AI is rough to make your data valuable.
4. Lines 584-590 need to be deleted. Citing other manuscripts in the conclusions is a mistake, if you want to include such information in the manuscript, do so in the discussion section.
5. Last but not least - the manuscript needs serious revision in terms of the language used. It's readable, but you need a professional service or an native English colleague to revise it.
In summary, I really like the data you collected and I admire your work, but I'd like you to sound a little less misleading. I think your manuscript aims to be a really good one. :-)
Author Response
Response to Reviewer 2
We thank the Reviewer for his/her positive evaluation of our manuscript and for helpful concerns to improve the article.
In this revised version of the work we have addressed the major concerns of the referee (the parts highlighted in green have been included in the manuscript).
Reviewer 2: I reviewed a manuscript examining the effects of a novel psychoactive compound in mice. NM2AI is pretty much dead, I think it literally died around 2016-2017 and it could be hard to purchase it on the internet, but considering that some compounds were already dead and then rose from their graves, I think it's worth publishing any data on it considering PubMed only has 4 articles on NM2AI. So, here's the list of my "buts":
Rev2Q1: Lines 21-22 - how is this even possible? The stuff is pretty much dead, I searched several forums and couldn't find any comments other than that NM2AI doesn't even come close to having the same effects as MDAI. I've also never seen a comparison of NM2AI to MDMA, and I study them daily. I've never seen an NM2AI post on an MDMA forum. I'd suggest changing the sentence to inform readers that the substance was developed as an analogue to MDAI, which dealers hope can work the same as MDMA.
AA: We thank the Reviewer 2 for pointing out this inaccuracy. We have inserted a more suitable version of the sentence in the specific section of the manuscript as suggested.
Rev2Q2: You can't use an EMCDDA report from 2015 to claim that NPS are on the rise. In fact, they aren't anymore. It's all about psychedelics nowadays, and NPS are currently a niche for "drug freaks". Please review this statement. They're a public health threat, but to say that they're constantly on the rise is an exaggeration. For example, over 200 tonnes of cocaine were seized in Europe in 2020 (official EMCDDA data). If you look at the latest data, you'll see that the numbers for NPS are constantly decreasing, while the "classic" drugs are increasing.
AA: We agree with Reviewer 2 about this point and thank him/her for pointing out this inaccuracy. We have re-written the specific introduction section about the topic to make it more clear.
Rev2Q3: 103-105 - You say that it's quite obvious that NM2AI is used at parties just like MDMA, and you quote UN data from 2015. All these statements need to go away. Just say it's an NPS, but right now it's not a big issue. However, since there is no scientific data available on NM2AI, you have decided to publish your manuscript now in case the substance resurfaces or for whatever reason. You don't have to try to prove NM2AI is rough to make your data valuable.
AA: We agree with Reviewer 2 about this point and thank him/her for pointing out this inaccuracy. We have re-written the specific introduction section about the topic to make it more clear.
Rev2Q4: Lines 584-590 need to be deleted. Citing other manuscripts in the conclusions is a mistake, if you want to include such information in the manuscript, do so in the discussion section.
AA: We agree with Reviewer 2 about this point and thank him/her for pointing out this inaccuracy. We have decided to delete such information and the relative references.
Rev2Q5: Last but not least - the manuscript needs serious revision in terms of the language used. It's readable, but you need a professional service or an native English colleague to revise it.
AA: We thank Reviewer 2 for his/her suggestion. An English revision of the manuscript was provided in this version.
Round 2
Reviewer 1 Report
Now the article can be published